# Biomimetic Self-Adhesive Structures for Wearable Sensors

**DOI:** 10.3390/bios12060431

**Published:** 2022-06-20

**Authors:** Feihu Chen, Liuyang Han, Ying Dong, Xiaohao Wang

**Affiliations:** Tsinghua Shenzhen International Graduate School, Tsinghua University, Shenzhen 518055, China; cfh19@mails.tsinghua.edu.cn (F.C.); hly21@mails.tsinghua.edu.cn (L.H.)

**Keywords:** biomimetic octopus, biomimetic mussel, wearable device, hydrogel, self-adhesion, pulsewave detection

## Abstract

Inspired by the adhesion ability of various organisms in nature, the research of biomimetic adhesion has shown a promising application prospect in fields such as manipulators, climbing robots and wearable medical devices. In order to achieve effective adhesion between human skin and a variety of wearable sensors, two natural creatures, octopus and mussel, were selected for bio-imitation in this paper. Through imitating the octopus sucker structure, a micro-cavity array with a large inner cavity and small outer cavity was designed. The fabrication was completed by double-layer adhesive photolithography and PDMS molding, and the adhesion capacity of the structure was further enhanced by the coating of thermal responsive hydrogel PNIPAM. The adhesive force of 3.91 N/cm^2^ was obtained in the range of the human body temperature. PDA-Lap-PAM hydrogel was prepared by combining mussel foot protein (Mfps) with nano-clay (Lap) as biomimetic mussel mucus. It was found that 0.02 g PDA-Lap-PAM hydrogel can obtain about 2.216 N adhesion, with good hydrophilicity. Through oxygen plasma surface treatment and functional silane surface modification, the fusion of the PDMS film with biomimetic octopus sucker structure and the biomimetic mussel mucus hydrogel patch was realized. The biomimetic octopus sucker structure was attached to the human skin surface to solve the problem of shape-preserving attachment, and the biomimetic mussel mucus hydrogel was attached to the sensor surface to solve the problem of sensor surface adaptation. The fusion structure was used to attach a rigid substrate piezoelectric sensor to the skin for a human pulsewave test. The results verified the self-adhesion feasibility of wearable sensors with biomimetic structures.

## 1. Introduction

There are many examples of using biomimetic structures to help human progress, such as to design stable airplane wings by imitating birds, to reduce the water resistance of a ship’s body by imitating a whale head, etc. People use biomimetic prototypes in nature to explore their functional principles and design models with similar structures to achieve different optical, mechanical and chemical properties. The preparation of biomimetic structures generally adheres to the following principles: study and imitate the structural strength characteristics, hydrophobic self-cleaning, self-perception and energy conversion principles of organisms in nature, develop and prepare new structures and materials, and provide support and assistance for the current social development and the progress and renewal of various equipment [1]. In recent years, with the development of micro and nanotechnology, biomimetic structures on a micro–nano scale have gradually shown greater application prospects and potential. At present, biomimetic micro–nano structures have achieved further application and development in industrial manufacturing, intelligent communication, health care, environmental protection and other aspects. For example, self-cleaning hydrophobic materials prepared by imitating a lotus leaf surface, flexible photosensitive discoloration leaves designed by imitating a chameleon, etc. In recent years, with the continuous improvement in people’s living quality and medical level, wearable medical monitoring equipment has gradually become popular. The biomimetic self-adhesion micro–nano structures can make functional devices that can automatically attach to the skin surface, transmit signals regarding the human body in fidelity, and realize long-term wearing and usage.

In the present studies, there are two types of natural surfaces with ultra-high adhesion: One has good adhesion to droplets, which is commonly referred to as the petal effect. Using natural petals such as red rose, clivia and sunflower as templates, polystyrene (PS) films [2], nanotube films [3] and other micro/nano structures [4,5] have been prepared, which show excellent hierarchical micro/nano structures similar to natural templates and have strong water adhesion. The other one shows a good dry adhesion function for various surfaces. Creatures such as geckos, octopuses and mussels can attach themselves to a wide variety of smooth, rough, wet and dry objects. These animals, through a variety of adhesion mechanisms, achieve an adhesion that can support much more than their body weight. 

The gecko’s feet are made up of numerous neatly arranged microscopic hairs called bristles (30–130 μm long and 5 μm in diameter) that further divide into hundreds of nanoscale knife-like structures (200–500 nm in diameter). When it comes to contact with any surface, the knife-like structure deforms, increasing the molecular contact area and converting the weak van der Waals forces into a powerful attraction that allows the gecko to climb up vertical walls. According to the adhesion mechanism of gecko’s feet, the artificial gecko bristles must have the following three basic characteristics:(1) high aspect ratio at a microscale (1:10–30) and nanoscale (1:20–50), (2) high density of micro-nano bristles, and (3) high hardness nano bristles to prevent mutual adhesion. The current biomimetic methods of gecko bristles are generally to prepare spiky column structures [6,7], column arrays with mushroom tips [8] or large-scale preparation using CVD [9], etc., but such a large aspect ratio and nanoscale secondary structure preparation of bristle ends cannot be obtained. 

The suckers of various sizes are one of the most important organs of octopuses, helping them to walk, adhere to and hunt in the complex marine environment [10]. Although a variety of biomimetic octopus sucker adhesion systems have achieved unique adhesion capabilities [11,12,13,14], most of them require certain external preloads to be applied during the adhesion process, or additional tools such as sucker systems and power sources to induce negative pressure in the inner cavity of the sucker. It is an important research direction for an intelligent adhesive system to control adhesion through external signals such as temperature, light, humidity and current. For the surface of human skin, it is a more suitable way to realize the biomimetic octopus sucker muscle contraction effect by temperature change. 

Mussels acquire the ability to stick to various surfaces in the ocean, such as rocks and ships, by secreting mussel foot proteins (Mfps) [15,16]. Mfps, including Mfp-2, Mfp-3, Mfp-4, Mfp-5 and Mfp-6, can form adhesive plaques between the foreign body surface and the substrate filaments of mussels, showing high interface binding strength, stability and toughness. Studies have found that these properties are due to the high content of catechol groups in Mfps. Adhesives formed by free radical polymerization of gelatin [17], polyethylene glycol [18] or dopamine methylacrylamide (DMA) [19] and other monomers are commonly used, but the oxide of the catechol group will have a certain slowing and inhibiting effect on polymerization. At present, the preparation method of biomimetic mussel hydrogel with a relatively simple technological process requires adhesion based on a dopamine (DA)-containing catechol group [20]. In the preparation process, the maintenance of an alkaline and oxidized environment needs to be further improved. Excessive oxidizing agent will destroy the degree of cross-linking, resulting in a decrease in the adhesive performance of the gel.

In general, the adhesion characteristics of the DA structures are not strong enough for the human skin surface, and their applications for various sensor surfaces are not enough either. Therefore, this paper selected two biomimetic adhesive micro–nano structures, namely, biomimetic octopus sucker structure and biomimetic mussel mucus structure, to study their respective preparation processes, and conduct performance tests on the prepared structures. The attachment problem of different sensors to human skin can be solved by optimization and fusion of the two biomimetic self-adhesive structures, and the feasibility of their applications in wearable sensors is verified by pulsewave test experiments.

## 2. Materials and Methods

### 2.1. The Biomimetic Octopus Sucker Structure

The process of preparing the biomimetic octopus sucker structure is as follows: Firstly, PR and LOR photoresist processes were performed to prepare the structure array mold [21]. Then, PDMS was used to obtain the array of cavity structures with a large inner cavity and a small outer cavity by demolding. Finally, a thermosensitive hydrogel PNIPAM was coated inside the cavity.

#### 2.1.1. Preparation of the Biomimetic Structure

Firstly, a 50 nm Ti metal film was coated on the surface of a 2-inch silicon wafer as a sacrificial layer (Figure 1a). After drying the silicon wafer, LOR 5B photoresist was spin coated at 500 RPM for 10 s, and a thickness of 0.6 μm [22] was obtained, then AZ5214 photoresist was spin coated at 2000 RPM for 30 s, and a thickness of 1.2 μm [23] was obtained (Figure 1b). For the LOR 5B film, soft baking of 60 s at 200 °C was followed after spin coating, and for the AZ5214 film, soft baking of 60 s at 95 °C was followed after spin coating. Then, 365 nm I line UV exposure for 5 s was performed (Figure 1c). After reverse baking at 110 °C for 60 s, 52 s flood exposure was performed (Figure 1d). Finally, the corresponding octopus sucker structure mold was obtained by 42 s developing with the specific developer for the positive photoresist (Figure 1e).

PDMS prepolymer and hardener were mixed in a mass ratio of 10:1 [24]. The level-1 speed of the homogenizer at 150 r/min for 15 s and, then, the level-2 speed at 300 r/min for 45 s were set to spin the mixture onto the silicon surface with the octopus sucker structure array mold (Figure 1f). After curing at 80 °C for 3 h in a vacuum oven, the silicon wafer was placed in Ti cleaning solution for full reaction to remove the underlying metal sacrificial layer, and then demolded and cleaned.

#### 2.1.2. Coating of the Thermosensitive Hydrogel

After the cavity structure array on the PDMS film was obtained, a thermosensitive hydrogel PNIPAM (Poly-N-isopropylacry-Lamide) was coated inside the cavity to achieve muscle contraction similar to the octopus sucker in the adsorption process, which can improve the adsorption force by increasing the internal and external pressure difference.

The properties of PNIPAM hydrogel are related to temperature. When the temperature is lower than the low critical solution temperature (LCST), the intermolecular force between the polymer chain and water molecules becomes strong enough to make the polymer chain linear and the water solubility increase, and the volume of the hydrogel increases through water absorption (swelling). However, when the temperature is higher than LCST, the interaction between the polymer and the solvent is weakened and the water solubility is reduced, leading to a decrease in hydrogel volume (desorption). Due to its safety and LCST in the skin temperature range, PNIPAM has been used for a variety of biomedical applications, including drug delivery, cell adhesion and joint lubrication in a variety of application scenarios [25,26].

The preparation process of PNIPAM is as follows: Firstly, 0.9 g NIPAM, 0.017 g BIS crosslinking agent and 0.025 g SDS accelerator were added into 60 mL deionized water and mixed thoroughly. Secondly, the beaker was sealed and filled with nitrogen for 30 min. Then, 5 mL APS solution with a concentration of 1.68 g/L was added to the mixture, and heated and stirred in a water bath at 65 °C for 4.5 h, during which nitrogen was continuously injected. PNIPAM was purified by deionized water dialysis for 7 days. The complete biomimetic structure was obtained by spin coating the PNIPAM onto the cavity of the prepared PDMS film. The cavity structure with PNIPAM is shown in Figure 1g.

### 2.2. The Biomimetic Mussel Mucus Hydrogel

The biomimetic mussel mucus PDA-Lap-PAM hydrogel was prepared by in situ polymerization. Its adhesion is attributed to the presence of sufficient free catechol groups in the hydrogel, which is formed by controlling the oxidation process of PDA in the closed nanolayer of nano-clay. The interlaminar structure of the nano-clay imitates the enclosed space in the mussel, and the catechol groups contained in the polymerized PDA have the same component groups as mussel foot protein. The specific preparation process is as follows: Firstly, 0.021 g DA was dissolved in 10 mL deionized water (DI) to form DA aqueous solution. Secondly, 0.26 g nano-clay (Laponite XLG) was added to the DA solution and stirred vigorously for 5 h to insert and oxidize DA to obtain the Lap–PDA mixture. Then, 2.6 g acrylamide (AM), 0.25 g ammonium persulfate (APS), 0.005 g N,N-methylene bisacrylamide (BIS) and 20 μL tetramethylenediamine (TMEDA) were added to the Lap–PDA suspension and stirred in an ice bath for 10 min. At last, the PDA-Lap-PAM hydrogel was obtained by curing at 40 °C for 1.5 h.

### 2.3. The Biomimetic Micro–Nano Fusion Structure

The process of fusing the two biomimetic structures is as follows: Firstly, 10 μL acetic acid and 2 g functional silane (methyl acryloxy propyl trimethoxysilane, TMSPMA) were added to 100 mL distilled water and stirred with magnetic stirrers for 10 min to fully mix [27]. Secondly, the PDMS film on the mold with octopus sucker structure array was washed with anhydrous ethanol and deionized water to remove surface dirt. Oxygen plasma was then bombarded with a plasma cleaner to form a primary hydrophilic surface of the PDMS film. The prepared silane mixture was poured into the petri dish containing the mold with the PDMS film and incubated at room temperature for 2 h to form a further hydrophilic surface of the PDMS film. Then, the silane solution was removed by anhydrous ethanol and the residual water was removed by nitrogen, and then the surface was allowed to dry. Finally, the unsolidified hydrogel precursor solution was poured into the Petri and solidified at the corresponding temperature for a certain period of time to obtain the biomimetic fusion structure with good adhesion. In the application process, the mussel mucus hydrogel was used as the upper glue to adhere to the sensor, and the biomimetic octopus sucker structure was used as the lower glue to adhere to the skin. Therefore, the biomimetic fusion structure realizes the adhesion connection between skin and sensor.

## 3. Results and Discussion

### 3.1. Preparation Result Characterization

To determine the successful preparation of the mold for the biomimetic octopus sucker structure, the completed structure on a silicon wafer was observed using a cold field emission scanning electron microscope. As shown in Figure 2a, columns with a larger radius formed by PR photoresist and columns with a smaller radius formed by LOR photoresist have complete structures, and the structure array has been successfully formed.

As shown in Figure 2b, a certain number of holes can be found in the SEM of the thermal response hydrogel PNIPAM. These pores are distributed on the sheet structure, which is intersected by many similar “branch-like” structures. Such a structure is conducive to the change in volume particle size through the absorption and release of water, so as to achieve the macroscopic results of thermal shrinkage and cold expansion. From the macro perspective, the appearance color of microgel at room temperature (25 °C) is a translucent color as shown in the left figure in Figure 2c. When the water bath is heated to 36 °C, the color changes to milky white [28]. These results indicate that the thermo-responsive hydrogel is successfully prepared and can achieve volume swelling-shrinkage change at 25–36 °C.

In order to verify the successful realization of free radical polymerization and PDA intercalation of the layered structure of nano-clay in the preparation process of the biomimetic mussel mucus hydrogel, X-ray diffraction (XRD) was used for characterization analysis [29]. For this purpose, the pure nano-clay Lap sample M0, PDA-Lap polymer M1, PDA-Lap-AM polymer M2 and PDA-Lap-PAM polymer M3 were prepared, respectively, representing each stage of hydrogel formation. For sample M0, nano-clay Lap was directly dispersed in 10 mL pure water and stirred vigorously for 5 h before freeze drying. For sample M1, DA was dissolved in deionized water to form DA aqueous solution, and then nano-clay Lap was added to the DA solution to make Lap-DA suspension. PDA-Lap polymer was obtained by stirring the suspension vigorously for 5 h. For sample M2, after the preparation step of M1, AM monomer was added and stirred in an ice bath for 10 min to obtain PDA-Lap-AM polymer, which was freeze-dried for backup. For sample M3, the full preparation process of the biomimetic mussel hydrogel was completed: After the previous preparation step of M2 polymer, APS and other initiators were added to promote the polymerization of AM to obtain the PDA-Lap-PAM polymer. According to the Bragg equation [30] and the distribution angle of the strongest peak in Figure 2d, the layer spacing of nano-clay Lap in the respective components of M0-M3 can be obtained. The results show that the layer spacing of nano-clay gradually increases with the progress of the reaction process. Thus, the biomimetic mussel mucus hydrogel was successfully prepared.

### 3.2. Adhesion

In the process of structural design and implementation, ensuring optimal adhesion is the most important consideration. In order to verify the adhesive ability of the biomimetic octopus sucker structure, the adhesive force between the biomimetic structure sample and glass sheet was tested using a push-tension gauge platform. Three different samples were prepared for comparison, and were tested in four groups according to the complexity of the structure. The first one is labeled as “No LOR” structure, for which, the cylindrical cavity structure was made from AZ5214 photoresist without the use of LOR at the bottom; The second is labeled as “LOR” structure, which has the octopus sucker with a large cavity and a small cavity by double-layer adhesive process, but is not coated by PNIPAM. The third is labeled as “PNIPAM”, which is the “LOR” structure coated with thermal response hydrogel PNIPAM, and according to different test temperatures, noted as “PNIPAM (25 °C)” at room temperature of 25 °C and “PNIPAM (40–25 °C)” at temperature of 25 °C when the temperature of the structure rises to 40 °C. As shown in Figure 3a, the experimental results show that compared with the pure cylindrical cavity (No LOR), the adhesive force of the biomimetic octopus sucker prepared by the double-layer adhesive is improved to different degrees, and the adhesion performance of the complete octopus sucker with PNIPAM is even improved to more than 4 times under the condition of heating up. The adhesive force of “PNIPAM(25 °C)” is not as good as that of “LOR”. However, when the temperature changes from room temperature to 40 °C, the adhesive force of the “PNIPAM (40–25 °C)” group was significantly improved, reaching 3.906 N/cm^2^, which was the optimal adhesive structure among the samples. The optimal adhesion structure was selected to conduct the cyclic repeatability test of adhesion-desorption, and the test results are shown in Figure 3b. During the 40 times of the repeated adhesion process, the adhesive force of the biomimetic octopus sucker structure remained stable at an average of 3.95 N/cm^2^, and there was no obvious attenuation of adhesive force. Therefore, the complete sucker structure with PNIPAM has good adhesion repeatability.

A total of 55 adhesion tests were carried out on the biomimetic mussel mucus hydrogels. During the test, 0.02 g hydrogel was taken to adhere between two glass sheets, and the adherence–desorption operation was repeated. As shown in Figure 4a, the results show that there is no loss of adhesive force in the 55 repeated tests, and the mussel mucus hydrogel has good adhesion repeatability, which is suitable for repeated use. The average adhesion force of the 1st to 25th tests is 2.0926 N, and the average adhesion force of the 26th to 55th tests is 2.3797 N. Taking the 25th test as the dividing line, the trend of adhesion force changed from lower to higher than the average adhesion force of 2.216 N. This result is caused by the increase in the preloading force from 7.5 N to 9 N, which proved that the increase in preloading force is beneficial to improve the adhesive force of biomimetic mussel mucus hydrogel.

After repeated tests, the same biomimetic mussel mucus hydrogel was used for 7 days of continuous adhesion, during which the adhesion was tested at 24 h intervals. The test results are shown in Figure 4b. The adhesive force did not attenuate in the first 5 days, but attenuated by 0.34 N and 0.44 N on the sixth and seventh days, respectively, compared with the first day, which was caused by the inevitable water loss in the long adhesion process of the hydrogel. The best use time is within 5 days after starting to attach. Biomimetic mussel mucus hydrogel has little effect on the overall adhesion during the continuous adhesion process of 7 days, which can meet the basic requirements of long-term use. When not in use, use ordinary plastic cling film to preserve the adhesive structure. The excellent adhesion repeatability and adhesion duration give the bionic adhesion structure good economic practicability. 

The universal tensile machine was used for the tensile test at a speed of 100 mm/min, as shown in Figure 4c. The initial length of the stretching was 3.3 cm. Except for the hydrogel with 13% Lap content that was broken, the lengths of the other two groups were all greater than 60 cm, and the stretching rate was over 1800%, showing excellent ductility. In addition, according to the slope of each curve, the 10% Lap group has the best elastic modulus and flexibility. Therefore, the hydrogel with 10% Lap content adopted in this paper has the best mechanical properties and meets the requirements of human skin application.

Biomimetic mussel mucus hydrogel shows good adhesion characteristics to various surfaces, which benefits from the mussel adhesion ability in nature. As shown in Figure 4d, a piece of PDA-Lap-PDA hydrogel was attached to the surface of human skin and adhered to the sports bracelet to achieve stable adhesion. In addition, the same piece of hydrogel was used to interconnect other surfaces, such as skin and glass bottles (Figure 4e), desktop side wall and a sports bracelet (Figure 4f), etc., and the surface adhesion was successfully achieved. The PDA-Lap-PDA hydrogel was pressed into a flat shape to adhere to the skin, sports bracelet and glass bottle. In addition, a trial test was carried out on the hydrogel used above. After repeated rubbing, the biomimetic mussel hydrogel was able to adhere to surfaces of the table (Figure 4g), leaves (Figure 4h) and uncleaned floors (Figure 4i), even if the impurities were mixed in. During the test, one end of the hydrogel sticks to various surfaces and a human hand pulls on the other end to stretch the gel into strips. It shows that hydrogel has good adhesion to complex unclean surfaces. Therefore, the PDA-Lap-PDA hydrogel has good adhesion adaptability with multiple surfaces.

### 3.3. Pulsewave Signal Acquisition

In order to verify the application of the biomimetic structures in wearable sensors, pulse waveform measurements were performed. A square area of 2 × 2 cm was marked on the left wrist radial artery area of a healthy 24-year-old male as the test region of the pulse sensor. In order to verify whether the pulsewave signal acquisition of the sensor with the biomimetic structure can achieve long-term monitoring, high-pressure detection and certain anti-disturbance ability, long-term heart rate detection, blood pressure measurement and arm disturbance pulse measurement were carried out. For each test, direct measurement without any adhesive and measurement with strong double-sided adhesive were used for comparison, as shown in Figure 5a. An inflatable wrist band was used to apply pressure, as shown in Figure 5b. A rigid piezoelectric sensor was used to verify the biomimetic fusion structure, as shown in Figure 5c.

#### 3.3.1. Heart Rate Measurement

Pulse waveform was monitored for 1 h by the rigid piezoelectric sensor, respectively, with non-adhesive, strong double-sided adhesive and the biomimetic fusion structure self-adhesion, and the wristband pressure was set to 20 mmHg.

Figure 6a is a partial schematic diagram of the measured pulse waveform near the 0th, 30th and 60th min under the conditions of direct measurement without adhesive (Figure 6a(i)), with double-sided adhesive (Figure 6a(ii)) and with the biomimetic fusion structure self-adhesion (Figure 6a(iii)). In terms of the waveform collection effect, the characteristic points of the main peak of the knock wave can be measured under the three test conditions and used to solve the heart rate and amplitude. However, the smoothness of the pulse waveform is the highest only under the condition of the biomimetic self-adhesion, where the waveform has no burr interference and retains the most complete isometric peak feature points. The characteristics of the secondary peak can be seen in the state of the double-sided adhesive, but there are still more burrs. Under the condition of direct measurement, the characteristic information of the secondary peak cannot be retained, and the noise interference is magnificent.

As shown in Figure 6b, under the three test conditions, the heart rate and amplitude obtained by solving the measured pulse waveform are compared in charts. The average heart rate obtained by the direct measurement without adhesive is 87 bpm, with double-sided adhesive 97 bpm, and with the biomimetic fusion structure self-adhesion 91 bpm. The average heart rates obtained by commercial electronic heart rate monitors were 88 bpm, 95 bpm and 90 bpm, respectively. If the subject stayed up late the day before the test, the upper limit of the overall heart rate was close to the normal value of 60–100 bpm, which is consistent with the physical state, and the heart rate signal can be obtained successfully in all three states. According to the longitudinal comparison of the amplitude, the average amplitude of the three test conditions is 25, 50.9 and 115.6, respectively. Under the same pressure condition of 20 mmHg, the biomimetic fusion structure self-adhesion can obtain the maximum pulse amplitude, which benefits from its excellent adhesion performance. The transverse comparison in the time dimension shows that the amplitude of the adhesive-free and the double-sided adhesive decreases by 40.7% and 20.4%, respectively, during the 1 h test, while the pulse amplitude of the biomimetic fusion structure self-adhesion increases by 41%. This is because when the biomimetic fusion structure is attached to human skin, the heat-responsive hydrogel begins to shrink, which reduces the internal pressure of the cavity and further increases the adhesion force, and the shape-preserving adhesion between the biomimetic structure and skin is gradually improved. However, for the non-adhesive situation, the adhesion will loosen during the measurement, reducing the quality of the collected waveform. For the double-sided adhesive situation, although the amplitude attenuation can be reduced due to the adhesion, the overall trend is still declining. Therefore, the biomimetic fusion structure contributes to a better performance of the rigid substrate sensor in long-term heart rate monitoring.

#### 3.3.2. Blood Pressure Measurement

Pulsewave oscillography was used to measure blood pressure [31]. In the test, the pressure of the wrist band was loaded up to 150 mmHg and then gradually lowered by a step of 10 mmHg. The stable waveform was guaranteed for at least 10 s each time until the pressure was reduced to 0 mmHg. The amplitude–pressure curve was drawn according to the amplitude of the waveform under each pressure, and the diastolic and systolic blood pressure were calculated according to the multiplier relationship.

Yellow circles and red circles were used to mark the peak and trough of the waveform obtained. Firstly, the waveform amplitude measured in the three adhesion conditions at 0 mmHg was compared, as shown in Figure 7. The amplitude of the waveform without adhesive is 6.37 (Figure 7a), and the waveform is chaotic and the basic characteristics of the pulse waveform are lost. The feature point is marked as the peak of clutter. Therefore, it is not feasible to directly measure pulse waveform under the condition of 0 mmHg. The amplitude of the waveform with double-sided adhesive is 24.62 (Figure 7b). The amplitude of the waveform is improved by the adhesion, but the secondary wave cannot be measured basically. The corresponding waveform amplitude at 0 mmHg pressure obtained with the biomimetic structure is 52 (Figure 7c), and the sub-wave characteristics have begun to appear. It further shows that the mechanical transfer characteristics of the biomimetic structure are optimal under low pressure.

Figure 8 shows the waveform in the condition of the biomimetic fusion structure adhesion at pressure loaded by the wristband decreasing gradually from 150 to 0 mmHg. The amplitude of the waveform in the figure shows a trend of increasing at first and then decreasing, which is consistent with the situation of the blood pressure test by the oscillography method. It can be seen from the local diagram that the pulse waveform meets the standard waveform at low pressure (0–110 mmHg), while the waveform at high pressure (120 mmHg and above) will have certain distortions due to the large extrusion pressure, but the basic amplitude information is retained. 

The amplitude–pressure curve can be obtained under the conditions of non-adhesive, double-sided adhesive and biomimetic fusion structure self-adhesion, as shown in Figure 9. The systolic and diastolic blood pressure of the human body were calculated according to the high and low pressure corresponding to 0.75 and 0.7 times the maximum amplitude A_M_. The diastolic/systolic blood pressure calculated from the amplitude–pressure curve of the biomimetic fusion structure self-adhesion was 72/111 mmHg, and the corresponding result of the standard electronic sphygmomanometer was 73/112 mmHg. The calculation of the non-adhesive test was 87/117 mmHg, and the corresponding standard result was 79/113 mmHg. The calculation of the double-sided adhesive test was 50/100 mmHg, and the corresponding standard result was 68/109 mmHg. The results of the three measurements showed a great error in the measurement of the double-sided adhesive. This is due to the excessive extrusion and deformation of the rigid cylindrical piezoelectric sensor on the double-sided adhesive at high pressure above 90 mmHg pressure, which reduced the pulse signal transmission efficiency. Moreover, at 120–140 mmHg pressure, adhesion jumped due to its poor ductility. This results in a decrease in the amplitude of the waveform. The biomimetic fusion structure itself with good adhesion properties and ductility can be gained under the condition of low pressure, which is better than that of the direct measurement without adhesive, and can reduce amplitude deviation by releasing the excessive squeeze of the rigid sensor to the skin and, therefore, can achieve good performance in terms of blood pressure measurement.

#### 3.3.3. Disturbance Test

In order to test the anti-disturbance ability of the biomimetic fusion structure, the pulse waveform was measured under the condition of arm swing. The pulse waveform of the subject was tested with the three types of attachment under the pressure of 20 mmHg of the wristband, while the subject swung his arm back and forth at an angle of 90° per second.

Figure 10 shows the pulse waveforms of the three attachment modes in an arm swing state. On the left is the result obtained without removing the baseline where the black line is the baseline, and on the right is the partial diagram after removing the baseline, and the peak and trough of the pulse waves are marked. It can be seen from the graph that the overall waveform baseline of the three attachment modes will drift with the disturbance caused by arm swing, but only the percussion wave generated in the mode of the biomimetic fusion structure self-adhesion can identify and follows the baseline drift well. In the states of non-adhesive and double-sided adhesive, the disturbance caused by the swinging arm has a great influence, and the feature points of the pulse waveform can hardly be identified in the graph on the right. However, the biomimetic fusion structure can still obtain clear waveform data after removing the baseline. Therefore, when the biomimetic fusion structure adheres to the skin surface for mechanical signal transmission, it has a certain tolerance to the disturbance caused by human movement.

## 4. Conclusions

Based on the self-adhesion mechanism of the biomimetic octopus sucker and mussel mucus, this paper prepared the biomimetic octopus sucker micro-cavity structure array, biomimetic mussel mucus hydrogel and the fusion structure of the two, providing a new idea for the design and implementation of biomimetic micro–nano adhesion structure. Combined with the characteristics of the heat-responsive hydrogel PNIPAM, the normal adhesion of the biomimetic octopus sucker was 3.91 N/cm^2^ in the human body temperature range. The biomimetic mussel hydrogel PDA-Lap-PAM was prepared by dopa (DA) and nano-clay (Lap). The adhesive force of 0.02 g PDA-Lap-PAM hydrogel was about 2.216 N. Both biomimetic adhesive structures have good adhesion repeatability. The biomimetic fusion structure adhesion was used to attach a rigid piezoelectric sensor to human skin for the acquisition of a pulse waveform and test of heart rate and blood pressure. The results demonstrate that it can realize a long time measurement, is resistant to high and low pressure, and has certain tolerance to body movement. These features promise the potential of the biomimetic fusion structure in the application of wearable sensors.

## Figures and Tables

**Figure 1 biosensors-12-00431-f001:**
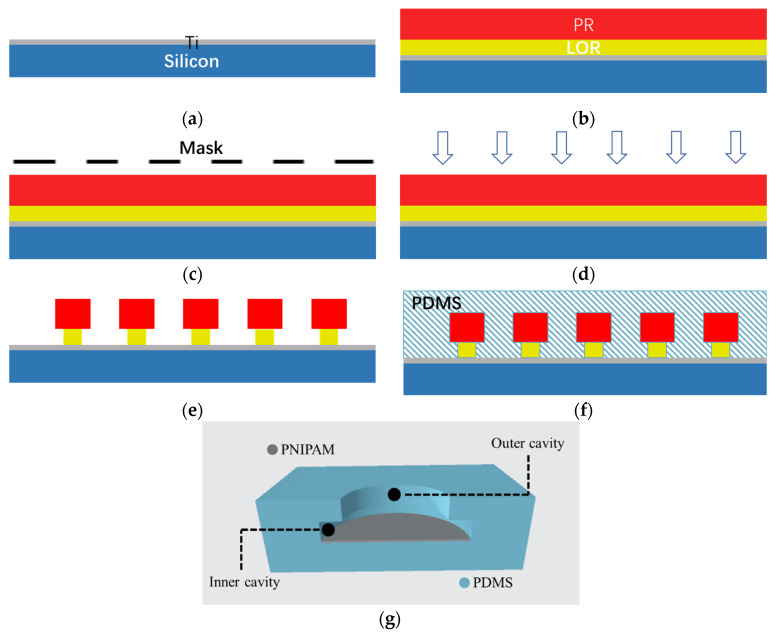
The biomimetic octopus sucker structure preparation process: (**a**) Silicon wafer coated with Ti film; (**b**) PR and LOR photoresist spin coating; (**c**) UV mask exposure; (**d**) UV flood exposure; (**e**) positive photoresist developing; (**f**) PDMS pouring onto the formwork; (**g**) Section view of the cavity structure after PDMS demolding and PNIPAM coating.

**Figure 2 biosensors-12-00431-f002:**
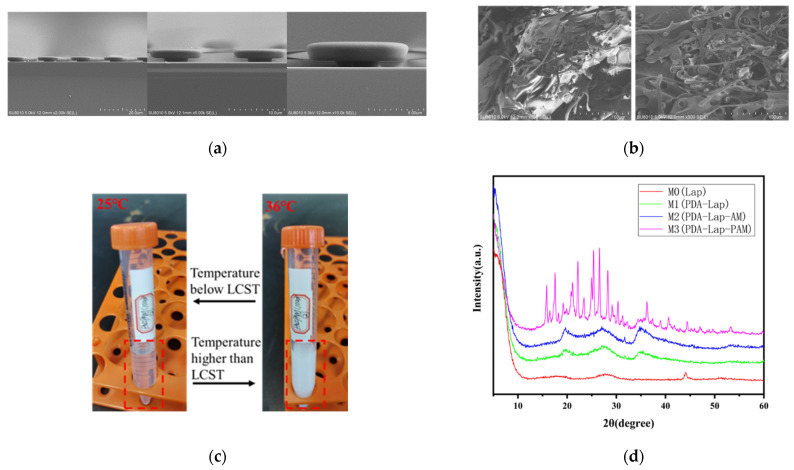
Preparation result characterizations of the two biomimetic structures. (**a**) SEM of the octopus sucker structure mold; (**b**) SEM of the thermosensitive hydrogel PNIPAM; (**c**) Color of PNIPAM varying with temperature; (**d**) XRD analysis diagram of the biomimetic mussel mucus hydrogel.

**Figure 3 biosensors-12-00431-f003:**
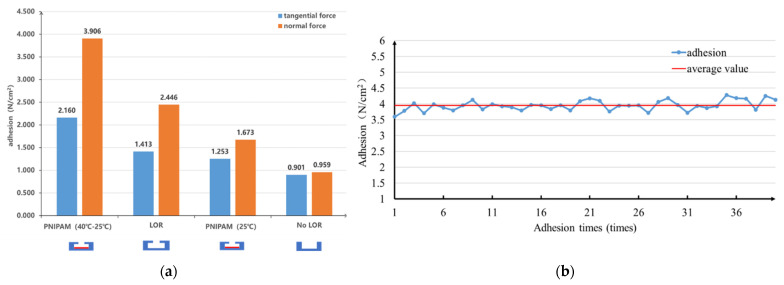
Test results of adhesive property of the biomimetic octopus sucker structure: (**a**) The normal force and tangential force of the four groups; (**b**) Repeated adhesion.

**Figure 4 biosensors-12-00431-f004:**
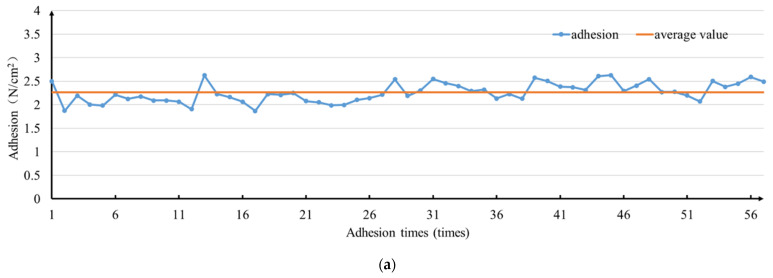
Test results of adhesive property of the biomimetic mussel mucus hydrogel: (**a**) Repeated adhesion; (**b**) Change in adhesion force for seven consecutive days; (**c**) Stress–strain curves of 7%, 10%, and 13% Lap content hydrogels; (**d**) As adhesive of skin surface and sports bracelet; (**e**) As adhesive of skin surface and glass bottle; (**f**) As adhesive of desktop side wall and sport bracelet; (**g**) Adhesion to desktop; (**h**) Adhesion to leaf surface; (**i**) Adhesion to uncleaned floor.

**Figure 5 biosensors-12-00431-f005:**
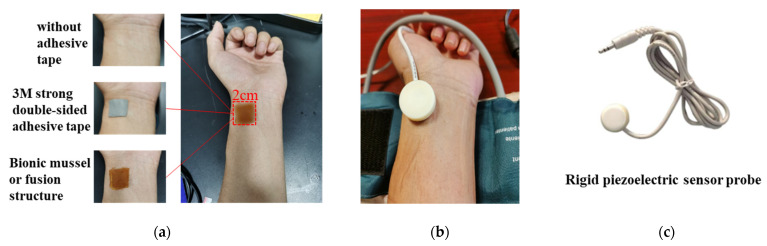
Pulsewave signal acquisition experiment setup: (**a**) Three different conditions of attachment; (**b**) The inflatable wrist band to apply pressure; (**c**) The rigid piezoelectric sensor for the test.

**Figure 6 biosensors-12-00431-f006:**
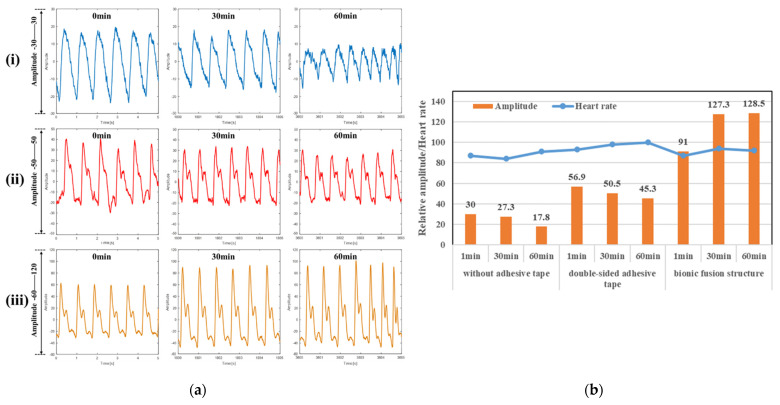
Pulse waveform and analysis of the rigid piezoelectric sensor at 0, 30 and 60 min under three kinds of attachment conditions: (**a**) The pulse waveform by the: i. direct measurement without adhesive; ii. with 3M strong double-sided adhesive; iii. with the biomimetic fusion structure self-adhesion; (**b**) Heart rate measurement results and amplitude contrast.

**Figure 7 biosensors-12-00431-f007:**
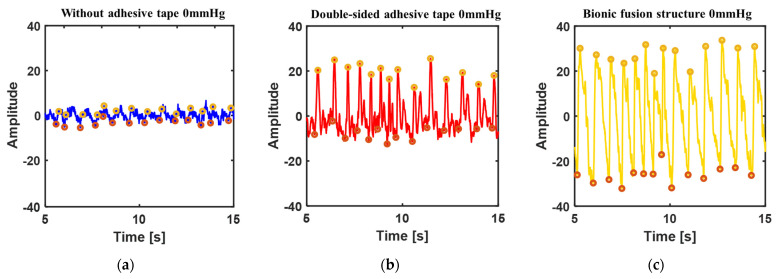
Comparison of pulse waveform under three adhesion conditions at 0 mmHg pressure: (**a**) Direct measurement without adhesive; (**b**) With 3M strong double-sided adhesive; (**c**) With the biomimetic fusion structure adhesion.

**Figure 8 biosensors-12-00431-f008:**
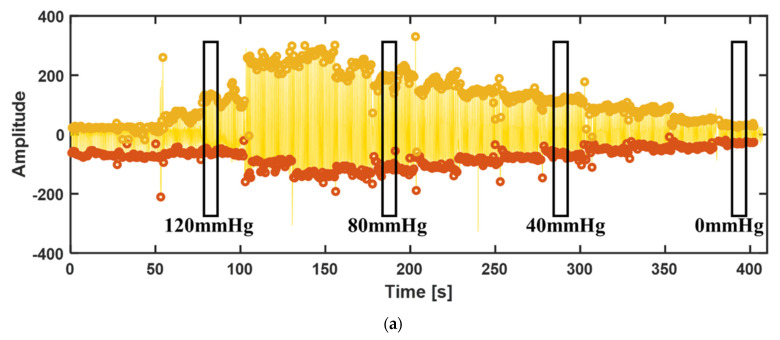
Pulse waveform with the biomimetic fusion structure adhesion at 150−0 mmHg pressure: (**a**) Global pulse waveform; (**b**) Pulse waveform at 40 mmHg pressure; (**c**) Pulse waveform at 80 mmHg pressure; (**b**) Pulse waveform at 120 mmHg pressure.

**Figure 9 biosensors-12-00431-f009:**
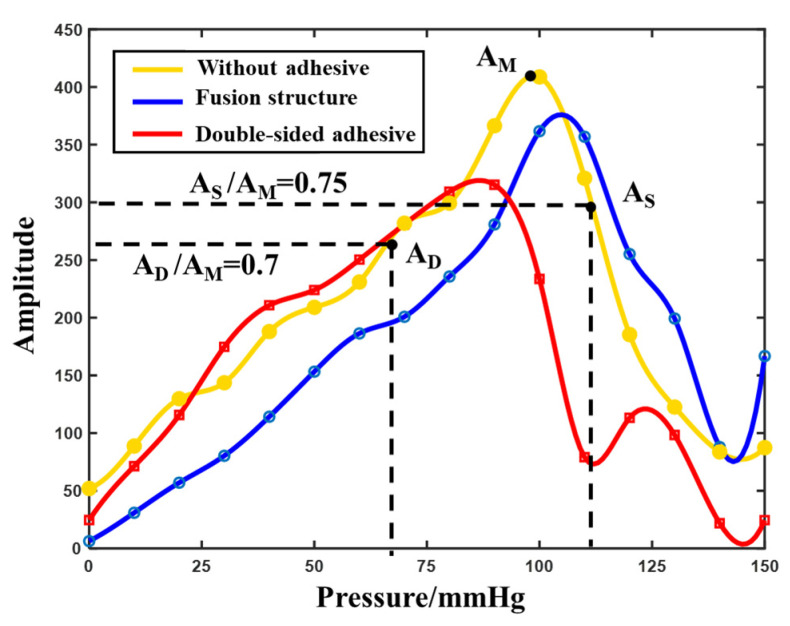
Diagram of blood pressure measurement by the oscillographic method under three attachment conditions.

**Figure 10 biosensors-12-00431-f010:**
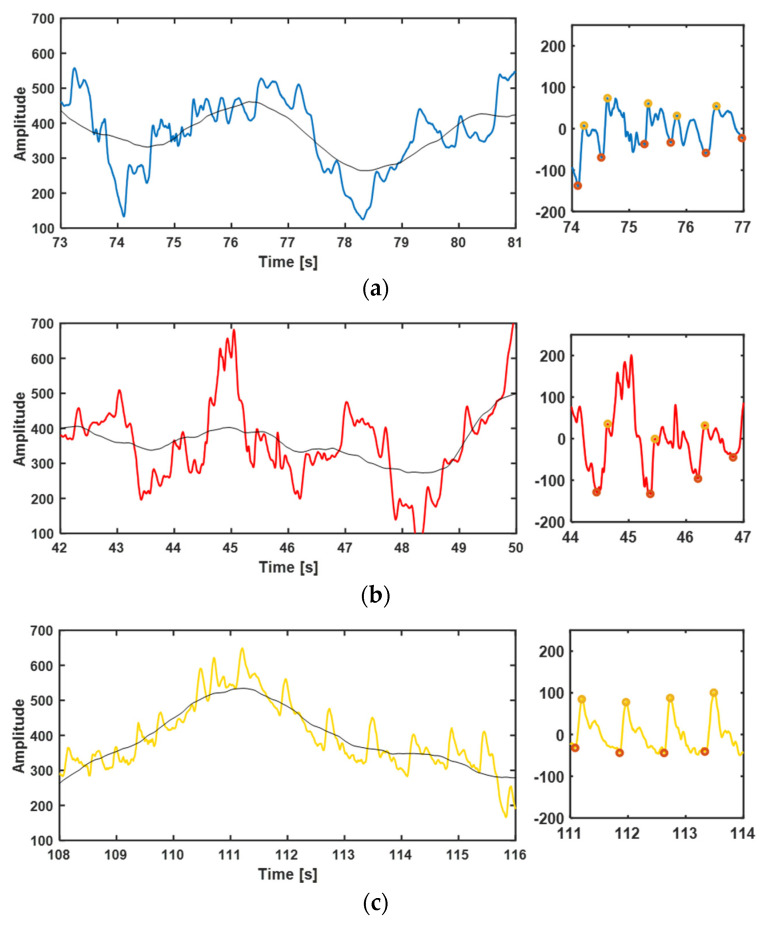
The waveform with disturbance due to arm swing under the three attachment conditions: Overall waveform with baseline (black line) (**left**) and local amplification waveform after removing the baseline (**right**) of (**a**) direct measurement without adhesive; (**b**) measurement with double-sided adhesive and (**c**) measurement with the biomimetic fusion structure self-adhesion.

## Data Availability

The data presented in this study are available on request from the corresponding author.

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
