# Peer review of "Biomimetic Self-Adhesive Structures for Wearable Sensors"

_biosensors, 2022, doi:10.3390/bios12060431_

Round 1

Reviewer 1 Report

This is interesting research and the data set was valuable. However, the manuscript needs further improvement and the following points need to be explained, answered, or modified:

In all the studied factors, which one is introduced as the main factor?

How is the sensor lifetime checked?

Did the authors pay attention to the economic aspect of the project? (For field use)

I propose explaining the construction method.

Does the hydrogel elasticity affect the sensor performance? The answer to the question is expected to be yes, in which case rheology tests are required.

Figure 4 is interesting for the reader. Please explain more about each part.

Reviewer 2 Report

The author in this paper (biosensors-1782276) reported a micro cavity array with large inner cavity and small outer cavity through imitating the octopus sucker structure to achieve effective adhesion between human skin and a variety of wearable sensors. The work is interesting and would be published after revisions:

1. The format of the manuscript should be checked, such as the space between the number and the unit.

2. The figures in the manuscript should be re-optimized to meet the standards of an academic paper.

3. In the “3.3.1. Heart rate measurement” section, the author should add the results of commercial device tests to verify the accuracy of the experimental results by comparison.
